# Magnetic field alignment of stable proton-conducting channels in an electrolyte membrane

Xin Liu [1], Yi Li[1], Jiandang Xue[1], Weikang Zhu[1], Junfeng Zhang [1], Yan Yin[1,2], Yanzhou Qin[1], Kui Jiao[1], Qing Du[1], Bowen Cheng[2], Xupin Zhuang[2], Jianxin Li[2] & Michael D. Guiver [1,3]

Proton exchange membranes with short-pathway through-plane orientated proton conductivity are highly desirable for use in proton exchange membrane fuel cells. Magnetic field is utilized to create oriented structure in proton exchange membranes. Previously, this has only been carried out by proton nonconductive metal oxide-based fillers. Here, under a strong magnetic field, a proton-conducting paramagnetic complex based on ferrocyanide-coordinated polymer and phosphotungstic acid is used to prepare composite membranes with highly conductive through-plane-aligned proton channels. Gratifyingly, this strategy simultaneously overcomes the high water-solubility of phosphotungstic acid in composite membranes, thereby preventing its leaching and the subsequent loss of membrane conductivity. The ferrocyanide groups in the coordinated polymer, via redox cycle, can continuously consume free radicals, thus helping to improve the long-term in situ membrane durability. The composite membranes exhibit outstanding proton conductivity, fuel cell performance and durability, compared with other types of hydrocarbon membranes and industry standard Nafion® 212.

[1] State Key Laboratory of Engines, Tianjin University, 300072 Tianjin, China. [2] State Key Laboratory of Separation Membranes and Membrane Processes, Tianjin Polytechnic University, 300387 Tianjin, China. [3] Collaborative Innovation Center of Chemical Science and Engineering (Tianjin), 300072 Tianjin, China. Correspondence and requests for materials should be addressed to Y.Y. (email: yanyin@tju.edu.cn) or to M.D.G. (email: michael.guiver@outlook.com)

Operational proton exchange membrane fuel cells (PEMFCs) rely on good proton transport from anode to cathode through proton exchange membranes (PEMs). Both perfluorocarbon-based PEMs (such as state-of-the-art Nafion®) and hydrocarbon-based PEMs typically exhibit isotropic conductivity or even unfavorable anisotropy with lower proton conductivity in the through-plane (TP) direction compared with the in-plane (IP) direction, despite often having distinct phase-separated morphology with tortuous-path conductive channels[1–8]. Ostensibly, a shorter and less tortuous conduction pathway in the TP direction would increase the efficiency of proton transport from the anode to the cathode, thereby increasing PEMFC performance. So far, efforts to orient conductivity in the TP direction have been explored by electric field alignment[9–15] and magnetic field alignment[16–20]. Electric field alignment has been explored by incorporating metal[9] or metal oxide[10], non-solvent-assisted casting[11], and polymer blends[12–15], but improvements in both conductivity and fuel cell performance have been modest. In the case of magnetic field alignment, this has been carried out using materials related only to metal oxides, such as pristine metal oxides[16–18], coated metal oxide[19], and other filler deposited with metal oxide[20]. This class of fillers for magnetic alignment are not proton conductive, and thus their incorporation would result in a reduction in ion exchange capacity (IEC), which compromises any gains in proton conductivity through enhancement of TP alignment. To circumvent this, highly proton conductive fillers that can be magnetically aligned are needed. Phosphotungstic acid (PWA) is a protonic Keggin-type polyoxometallate (POM) with a stoichiometry of $H_3PW_{12}O_{40}$. It has been used as a PEM filler because it is a strong protic acid with high thermal stability[21,22]. While PWA itself is not susceptible to magnetic field alignment because it is diamagnetic, a paramagnetic compound formed by the PWA Keggin POM anion and an electron donor material has been reported[23], which has the potential to be directionally aligned.

A well-recognized obstacle to using PWA composite PEMs is the high water solubility of PWA[24,25], and its progressive leakage from the membrane during use results in a deterioration of proton conductivity. To date, several strategies have attempted to stabilize PWA in PEMs, such as conversion of PWA into a water-insoluble form by partially substituting the proton on PWA with $Cs^+$ or $NH_4^+$ ions[26–28]. Other approaches involve immobilization of PWA onto water-insoluble supports, such as $SiO_2$[29,30], $ZrO_2$[31,32], mesoporous silica[33,34], and carbon nanotube[35]. More recently, amino-containing polymers, stabilized PWAs by hydrogen bonding[36], electrostatic force[37,38], or acid–base interactions[36,39] have been reported, as well as chemical bonding of lacunary silicotungstic acid (SiWA, similar to PWA) to polymers[40–42]. With these approaches, enhanced stability of PWA composite PEMs was achieved, but long-term membrane stability tests (>30 days) under severe conditions (>90 °C in water) are not reported.

Apart from the commonly observed problem of heteropoly acid filler leakage from composite PEMs, the overall durability of PEMs, including chemical, mechanical, and thermal stabilities during in situ operation are crucial to adequate PEMFC lifetimes[43,44]. Here, chemical stability refers to the endurance of PEMs toward radical attack (mainly OH• and OOH•). The most widely used strategy to improve PEM chemical stability is to incorporate free radical decomposition catalysts based on transition metals. While oxides of Mn and Ce ions have been shown to mitigate free radicals[45–47], Fe and Cu ions are well-known chemical degradation accelerators[48,49]. Other studies incorporated small molecular antioxidants, including Vitamin E[50], antioxidant 1010[51], and dihydroxy-cinnamic acid[52]. Among various heteropoly acids, doped SiWA[40] have been reported to be

relatively stable to radical degradation, but there appears to be less understanding on approaches to further enhance PEM chemical durability. Both OH• and OOH• have unpaired electrons and are highly electrophilic[53,54], so we advance the hypothesis that electron-rich regions of the PEM structure are more susceptible to OH• and OOH• attack. Lending support to this is that carboxyl, sulfonic, or ether groups are usually more sensitive to OH• and OOH•[55,56]. Accordingly, PEMs incorporating catalytic species with strongly negatively charged groups would be an effective way to mitigate against in situ chemical degradation.

To simultaneously realize the above-mentioned concept of TP alignment, stable PWA incorporation and membrane in situ durability, we synthesize an electron-donating, proton-conducting and redox polymer, ferrocyanide-coordinated poly (4-vinylpyridine) (CP4VP). Its electron-donating ability is due to cyano ligands with lone pair electrons, which may form a paramagnetic complex with PWA. This complex would be effective in realizing both TP magnetic alignment and PWA retention, and concurrently maintain proton conductivity, as both CP4VP and PWA are proton conductive. The redox reaction of CP4VP derives from transitions between ferrocyanide (Fe(II)) and ferricyanide (Fe(III))[57,58]. Due to the strongly negative charge on this redox couple, it would be capable of continuously consuming electrophilic OH• and OOH• radicals during the redox process, thus helping to shield other regions of the PEM structure. An additional advantage of this approach is that the redox groups are tethered to the polymer, so they do not readily migrate and aggregate like previous strategies using ions or small molecules, especially after forming a complex with PWA. In the present study, via magnetic-assisted solution casting, a composite PEM containing TP-aligned proton channels with retentive PWA is prepared. CP4VP and PWA are used as the proton-conducting components (PCs) and both are individually water soluble. The PCs are co-cast with a non-conductive polymer, polysulfone (PSf), to afford mechanical strength for the membrane. Under the inducement of a strong magnetic field, the two water-soluble PCs, CP4VP and PWA, combine to form a water-insoluble paramagnetic complex via electron transfer, which is synchronously oriented in the TP direction by the magnetic field. With these aligned and durable short-pathway proton transport paths, the composite membrane displays outstanding TP proton conductivity, with a value of 215 mS cm$^{-1}$ at 95 °C in water, with strong stability for a period of 30 days without degradation. The single PEMFC shows a high maximum power density up to 1107 mW cm$^{-2}$ at 95 °C and 100% relative humidity (RH), which is 172% of that for the PEMFC with Nafion® 212. In PEMFC durability tests, it exhibits excellent endurance, showing a small decline in current density of 4.1% over 32 days, compared with a 40% decline after 20 days for Nafion® 212, suggesting it could be a viable approach for PEMFC application.

## Results

**Synthesis and characterization of coordinated polymer.** The synthesis of CP4VP is shown in Fig. 1. Poly(4-vinylpyridine) (P4VP) is dissolved in methanol, and sodium pentacyanoammineferroate (II) (SPCAF) together with 1,4,7,10,13-pentaoxacyclopentadecane (15-crown-5) are dissolved in water. After mixing the two reactant solutions, 15-crown-5 acts as a phase transfer agent to chelate the $Na^+$ in SPCAF and transfer SPCAF from the water to methanol phase to react with P4VP. Then, excess HCl is used to protonate the coordinating group of CP4VP, giving the final green-yellow product. Unless specifically mentioned, CP4VP refers to fully acidified form hereinafter. Each CP4VP repeat unit has active protons and lone electron pairs on the cyano ligands, allowing CP4VP to be both proton conductive

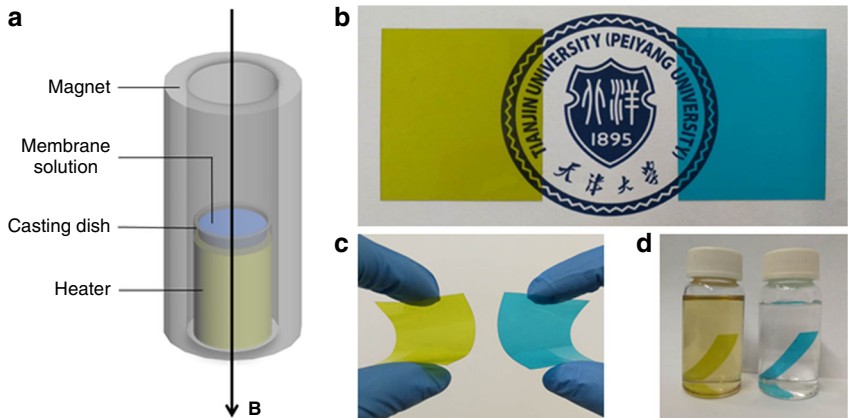

**Fig. 1** Synthesis of coordinated polymer. CP4VP is synthesized from P4VP and SPCAF. The pyridyl group in P4VP replaces the –NH₃ ligand of Fe and becomes a substitutive ligand to obtain the coordinated polymer. CP4VP: ferrocyanide-coordinated poly(4-vinylpyridine); P4VP: poly(4-vinylpyridine); SPCAF: sodium pentacyanoammineferroate (II); 15-crown-5: 1,4,7,10,13-pentaoxacyclopentadecane

**Fig. 2** Fabrication and features of composite membrane. **a** Diagram of the magnetic-assisted solution casting. Photos for NM-45PC (left in each subfigure) and MM-45PC (right in each subfigure) to show their appearances. **b** Both membranes are visibly homogeneous and transparent. **c** Both membranes are flexible due to the tough PSf matrix. **d** NM-45PC membrane shows leaching of the colored PC component CP4VP in water, but MM-45PC is insoluble. NM-45PC: normal-cast membrane with 45 wt% loading of proton-conducting components; MM-45PC: magnetic-cast membrane with 45 wt% loading of proton-conducting components; PSf: polysulfone; PC: proton-conducting components; CP4VP: ferrocyanide-coordinated poly(4-vinylpyridine)

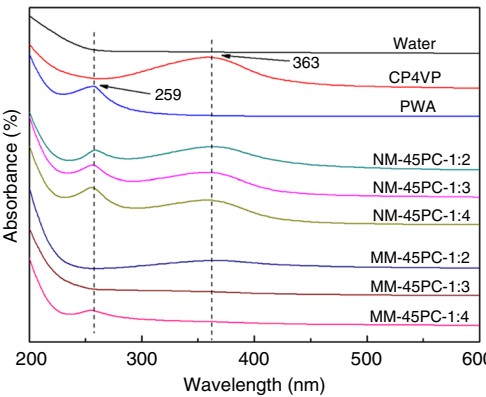

**Fig. 3** Ultraviolet–visible spectral analysis of leach water for various membranes. Among all composite membranes with various CP4VP/PWA ratios, only MM-45PC-1:3 shows no PC leakage (dashed lines are used to guide the eyes). CP4VP: ferrocyanide-coordinated poly(4-vinylpyridine); PWA: phosphotungstic acid; NM-45PC: normal-cast membrane with 45 wt% loading of proton-conducting components; MM-45PC: magnetic-cast membrane with 45 wt% loading of proton-conducting components; PC: proton-conducting components. Source data are provided as a Source Data file

and serve as an electron donor to combine with PWA Keggin POM anions. The ferrocyanide–ferricyanide redox couple is capable of continuously consuming free radicals, thereby mitigating membrane chemical degradation. In Supplementary Note 1, characterization of 100% functionalized (coordinating group) and acidified (H⁺) CP4VP by X-ray photoelectron spectroscopy (XPS), Fourier transform infrared spectroscopy (FTIR), thermogravimetric analysis (TGA), differential scanning calorimetry (DSC), IEC, and elemental analysis are provided as Supplementary Figure 1 and Supplementary Table 1.

**Fabrication and characterization of composite membrane.** The oriented CP4VP/PWA/PSf composite membrane is fabricated by solution casting under magnetic field alignment. As shown in Fig. 2a, the membrane solution is decanted into a casting dish, which is horizontally placed with a heater in the upright water-cooled magnet well to evaporate the solvent, with a 35 tesla field exerted. The membrane cast with the magnetic field is hence referred to as magnetic-cast membrane (MM). Comparative normal-cast membrane (NM) without magnetic field is to verify the influence of the magnetic field on membrane formation, more specifically, in order to clarify the magnetic facilitated interaction

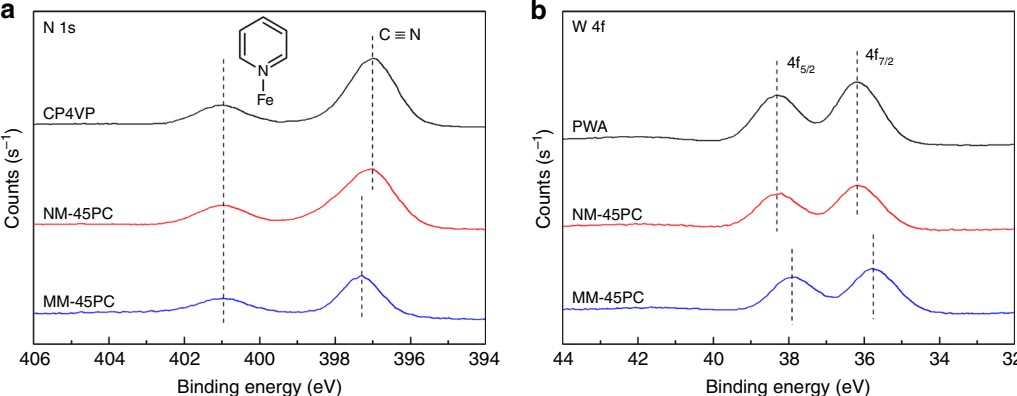

**Fig. 4 Comparative X-ray photoelectron spectra.** XPS narrow scans of N 1s and W 4f binding energy peaks for MM-45PC relative to membrane composition and NM-45PC are shown. **a** The N 1s scan of MM-45PC shows a positive shift for the cyano group relative to CP4VP and NM-45PC, indicative of a decreased electron density on the nitrogen of the cyano group. **b** The W 4f scan of MM-45PC shows negative shifts for $4f_{7/2}$ and $4f_{5/2}$, indicative of the increased electron density on the PWA tungsten atoms. In combination, these shifts in binding energies demonstrate that under magnetic field influence, electron transfer and an interaction occur between CP4VP and PWA in MM-45PC (dashed lines are used to guide the eyes). XPS: X-ray photoelectron spectroscopy; CP4VP: ferrocyanide-coordinated poly(4-vinylpyridine); PWA: phosphotungstic acid; NM-45PC: normal-cast membrane with 45 wt% loading of proton-conducting components; MM-45PC: magnetic-cast membrane with 45 wt% loading of proton-conducting components. Source data are provided as a Source Data file

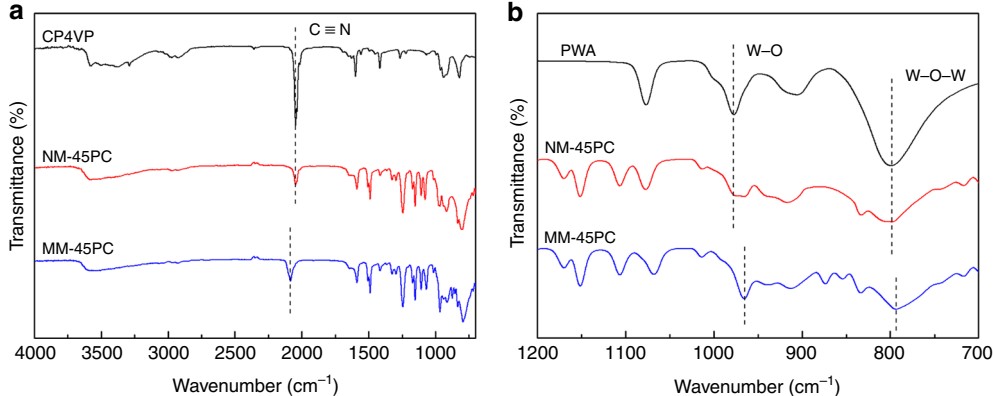

**Fig. 5 Comparative infrared spectra.** ATR-FTIR spectra of MM-45PC relative to membrane composition and NM-45PC are shown. **a** MM-45PC shows red shift for the cyano group absorption band, indicating an enhanced vibration of C≡N bond due to the loss of the electron on nitrogen. **b** MM-45PC shows blue shifts for W–O and W–O–W vibrations, reflecting attenuated vibrations of W–O and W–O–W bonds caused by the transfer of electrons from nitrogen to tungsten. In combination, these shifts in absorption bands demonstrate that under magnetic field influence, electron transfer and an interaction occur between CP4VP and PWA in MM-45PC (dashed lines are used to guide the eyes). ATR-FTIR: attenuated total reflection flourier transformed infrared spectroscopy; CP4VP: ferrocyanide-coordinated poly(4-vinylpyridine); PWA: phosphotungstic acid; NM-45PC: normal-cast membrane with 45 wt% loading of proton-conducting components; MM-45PC: magnetic-cast membrane with 45 wt% loading of proton-conducting components. Source data are provided as a Source Data file

between CP4VP and PWA, together with the TP alignment and formation of proton-conducting channels.

To ensure the stability of MM, the two water-soluble PC components, CP4VP and PWA, should have a carefully matched ratio so that neither is in excess. An optimized wt% ratio of 1:3 for CP4VP/PWA is determined after a series of experiments, discussed in detail later. If not specifically mentioned hereinafter, NM and MM refer to membranes with a CP4VP/PWA ratio of 1:3. NM-45PC and MM-45PC (the NM and MM with 45 wt% PC loading) are chosen for detailed investigation as 45 wt% is the highest PC loading available, which shows the highest performance and the most obvious contrast. When PC loading is higher than 45 wt%, both NM and MM become less transparent and mechanically weak.

After solution casting, both NM-45PC (left images in Fig. 2b–d) and MM-45PC (right images in Fig. 2b–d) are homogeneous, transparent, and flexible membranes, but they show two

important differences. First, NM-45PC retains the green-yellow coloration of CP4VP, which is the only original colored material in the membrane formulation, but MM-45PC acquires a blue coloration (right images in Fig. 2b–d). Interestingly, the membrane solution assumes a blue coloration during the magnetic-assisted solution casting process. Second, NM-45PC is partially soluble when acidified or washed with water, while MM-45PC remains insoluble (Fig. 2d).

To determine the most stable membrane formulation from the CP4VP/PWA ratio (optimized ratio determined to be 1:3), together with the water solubility of NM-45PC and MM-45PC, the membranes are soaked in water which is analyzed by ultraviolet–visible (UV–Vis) spectroscopy, as shown in Fig. 3. Pure water shows no absorbance, and solutions of CP4VP and PWA display absorbances at 363 and 259 nm, respectively. NM-45PC with CP4VP/PWA ratios of 1:2, 1:3, and 1:4 show absorbances at 363 and 259 nm, indicating both CP4VP and

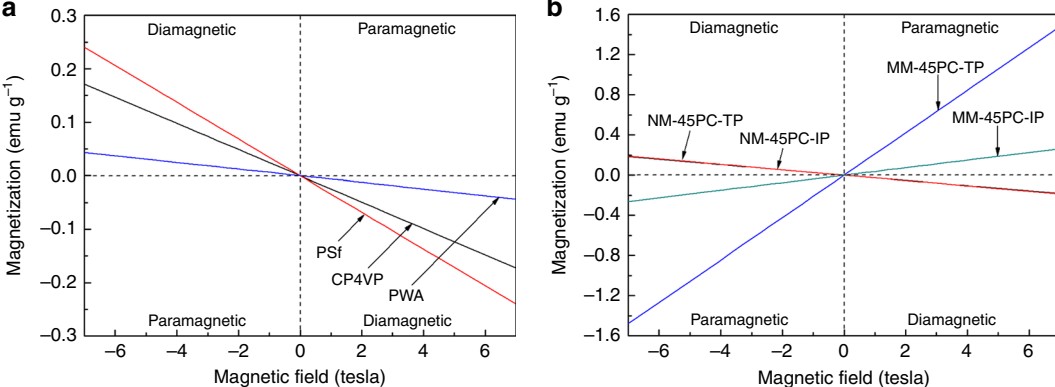

**Fig. 6** Magnetometry analysis. VSM measurement to determine the magnetism of individual membrane formulation components and membranes is shown. **a** Membrane formulation components CP4VP, PWA, and PSf display diamagnetism, and cannot be magnetically aligned. **b** Membrane NM-45PC is diamagnetic, but MM-45PC has paramagnetic properties in both TP and IP directions due to the formation of a paramagnetic CP4VP/PWA complex under magnetic field. The much higher $\chi$ value in TP (Supplementary Table 2) indicates that the paramagnetic complex is synchronously oriented in the TP direction during magnetic field solution casting (dashed lines are used to guide the eyes). VSM: vibrating sample magnetometer; CP4VP: ferrocyanide-coordinated poly(4-vinylpyridine); PWA: phosphotungstic acid; PSf: polysulfone; NM-45PC: normal-cast membrane with 45 wt% loading of proton-conducting components; MM-45PC: magnetic-cast membrane with 45 wt% loading of proton-conducting components; TP: through-plane; IP: in-plane. Source data are provided as a Source Data file

PWA leach out from the NMs. In contrast, MM-45PC with a CP4VP/PWA ratio of 1:2 shows a slight CP4VP absorbance, while a ratio of 1:4 shows a slight PWA absorbance. Only MM-45PC-1:3 shows no extraneous absorbance, indicating that neither CP4VP nor PWA leach out of the PSf matrix when the CP4VP/PWA ratio is 1:3.

Insight into differences in color and solubility between NM-45PC and MM-45PC are shown by XPS in Fig. 4. NM-45PC and CP4VP share the same N 1s peak binding energies for pyridine-Fe and cyano groups in Fig. 4a. However, MM-45PC shows a positive shift for the N 1s binding energy of the cyano group, which is attributed to a reduction in electron density, with the N 1s signal of the pyridine-Fe unchanged. Concurrently, negative shifts are observed in Fig. 4b for the W 4f binding energy peaks of MM-45PC, compared with those of PWA and NM-45PC, reflecting increased electron density. These structural differences between NM-45PC and MM-45PC are also evident in the comparative attenuated total reflectance FTIR (ATR-FTIR) spectra shown in Fig. 5a, for absorption bands related to the cyanide group and W–O/W–O–W, where alterations in chemical bond vibrations also derive from variations in electron density. These results indicate that, under the influence of the magnetic field, an interaction is built in MM-45PC between CP4VP and PWA via electron transfer from the cyano donor to the PWA acceptor. Since the XPS curves of W and the FTIR absorption bands for W–O/W–O–W exhibit no peak-differentiating phenomena, the transferred electrons should be delocalized over the PWA Keggin POM, that is, hopping rapidly among different W atoms, which is in good agreement with previous research[23,59]. Accordingly, the resulting CP4VP–PWA complexes are paramagnetic due to the delocalized unpaired electron over the Keggin-type POM[23,59], thus having the potential to be magnetically aligned. The reduced Keggin POM anion becomes blue when receiving transferred electrons, which is a known phenomenon and is termed "heteropoly blue" in the literature[60–62]. Most significantly, it is worth noting that heteropoly blue formation has previously been reported only by chemical, electrochemical, and photochemical methods, but here we report the magnetic inducement route. With the magnetic facilitated interaction,

CP4VP and PWA form a water-insoluble combination to produce a highly stable PWA composite PEM.

Vibrating sample magnetometer (VSM) tests are carried out at 80 °C (the same temperature used as for membrane casting) to examine membrane conducting channel alignment under magnetic field. Magnetic susceptibility ($\chi$) is calculated from VSM data and values are listed in Supplementary Table 2. The VSM data of individual membrane formulation components CP4VP, PWA, and PSf in Fig. 6a show that the magnetization lines all appear in the second and fourth quadrants, indicating each of the materials is diamagnetic and $\chi$s is negative. For NM-45PC and MM-45PC, VSM tests are implemented in both TP and IP directions. Figure 6b reveals that both TP and IP directions of NM-45PC show diamagnetic behavior, and their VSM lines are almost superimposed, indicating that NM-45PC has isotropic magnetism. In sharp contrast, both TP and IP of MM-45PC are paramagnetic, indicating that after electron transfer from CP4VP to PWA, the formed CP4VP/PWA coordination complex possesses paramagnetism, which is supported by similar findings in the literature[21]. Moreover, a much steeper slope in the TP direction reveals that the influence of the magnetic field during membrane casting generates not only electron transfer to obtain paramagnetic membrane properties but also endows MM-45PC with a strongly anisotropic orientation in the TP direction.

Clear visual evidence for the isotropy of NM-45PC and the TP orientation in MM-45PC are provided by transmission electron microscopy (TEM) observations of cross-section membrane images. The NM-45PC membrane exhibits a non-aligned distribution of black dots in Fig. 7a, which are believed to be PWA particles, since heavy atomic nuclei like tungsten are more electron opaque. Dynamic light scattering (DLS) indicates that the PWA particle size is ~1.26 nm, as shown in Fig. 7b, which is in good accordance with the TEM observation. The NM-45PC membrane presents significant agglomeration of PWA particles, which is a commonly observed phenomenon in high loadings of nanoparticle-filled composite membranes. In sharp contrast, MM-45PC displays directionally aligned evenly spaced black dots along the path of the magnetic field in Fig. 7c. Moreover, the dark-colored PWA particles appear to be separated by lighter-colored halo-like thin layers, such that improved dispersion

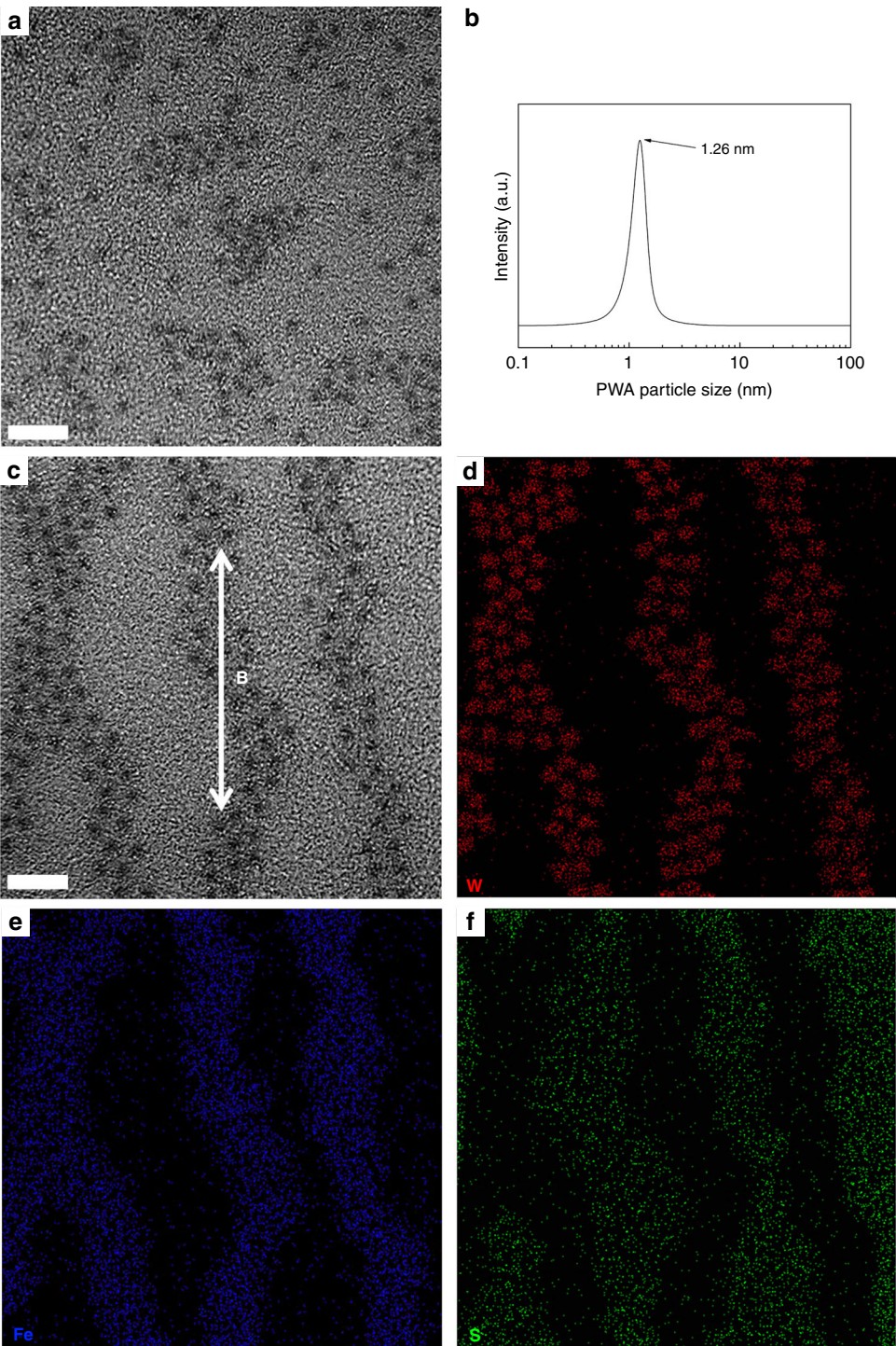

**Fig. 7** Membrane cross-sectional images, mappings, and filler particle size estimation. **a** TEM image of NM-45PC showing a random dispersion and agglomeration of PWA particles (5 nm scale bar). **b** PWA particle size (dispersed in benzene) estimation by DLS technique. The PWA nanoparticle diameter of 1.26 nm is in good accordance with size estimation from the TEM images. **c** TEM image for MM-45PC (5 nm scale bar). Directionally orientated chains of PWA particles are observed, and the halo-like areas located around the PWA particles are CP4VP domains. The unpopulated regions correspond to PSf domains. **d** Tungsten (W) mapping for the same region in **c**. **e** Iron (Fe) mapping for the same region in **c**. **f** Sulfur (S) mapping for the same region in **c**. TEM: transmission electron microscopy; EELS: electron energy loss spectroscopy; DLS: dynamic light scattering; CP4VP: ferrocyanide-coordinated poly(4-vinylpyridine); PWA: phosphotungstic acid; PSf: polysulfone; NM-45PC: normal-cast membrane with 45 wt% loading of proton-conducting components; MM-45PC: magnetic-cast membrane with 45 wt% loading of proton-conducting components. Source data are provided as a Source Data file

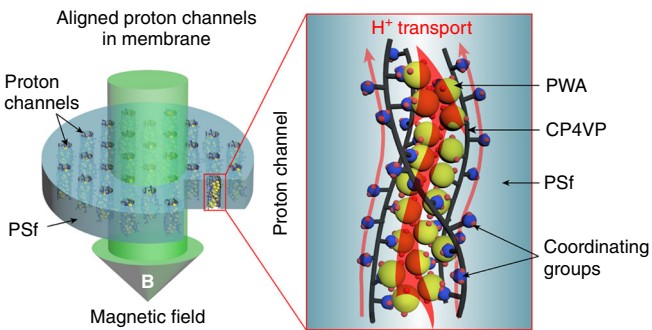

**Fig. 8** Conceptual diagrams. Magnetically aligned composite membrane and proton transport in the aligned channels are shown. CP4VP and PWA, both proton-conducting components, form TP-oriented proton channels, allowing proton transport along the coordinating ferrocyanide group/PWA complex and the uncombined coordinating ferrocyanide groups. CP4VP: ferrocyanide-coordinated poly(4-vinylpyridine); PWA: phosphotungstic acid; PSf: polysulfone; TP: through-plane

without agglomeration is evident. Based on the evidence for the formation of a complex between CP4VP and PWA under magnetic field, the lighter areas immediately surrounding PWA particles are deduced to be CP4VP domains, while the remaining brightest areas are inferred to be PSf membrane matrix. These inferences are verified by electron energy loss spectroscopy (EELS) mapping of W, Fe, and S for the same region shown in Fig. 7c, which are displayed in Fig. 7d–f, respectively. Figure 7d shows that tungsten atoms, present only in PWA of the membrane formulation, correspond to the black dots in the aligned chains, corroborating the inference that these dots are PWA. Figure 7e shows that iron atoms, present only in CP4VP, are located intensively on the black dots and associated halo-like regions, which confirms that CP4VP and PWA are closely interlinked. As the unique element in PSf, sulfur in Fig. 7f presents mainly in the brightest areas corresponding to Fig. 7c, identical with regions unpopulated by CP4VP and PWA. The combined results of TEM and EELS mapping demonstrate that the CP4VP/PWA complex in MM-45PC forms channel-like phases along the magnetic field, and PSf is a separate phase in which the channels are embedded.

Integrating the findings of the VSM and TEM analyses, both the formation and alignment of PC channels in MM-45PC are established concurrently under magnetic field during membrane casting. First, electron transfer from CP4VP to PWA is stimulated under the strong magnetic field, generating paramagnetic PC complex. Second, the paramagnetic complex, attached directly to the CP4VP polymer chain, is mobile in the membrane casting solution, allowing alignment along the direction of the magnetic field. Thus, a composite PEM containing TP-aligned proton channels composed of complexed retentively stable PWA is successfully fabricated, as depicted in Fig. 8. The high efficiency of proton transport in the aligned proton channels occur both through the coordinating group/PWA combination and the uncombined coordinating group. In summary, utilizing a highly proton-conducting filler, PWA, tethered to a proton conductive polymer (CP4VP), we prepare a PEM with TP proton-conducting channels by magnetic field alignment. Previous research has only used proton non-conductive fillers, such as metal oxides.

**Membrane properties**. Since the conducting portion of the NM-45PC membrane is water soluble, only the magnetically aligned MM-45PC membrane is evaluated and referenced against state-of-the-art Nafion® 212. The electron conductivity of MM-45PC membrane is experimentally measured and found to be negligible (Supplementary Note 2 and Supplementary Figure 2). Figure 9a shows that the proton conductivity of Nafion® 212 in the IP direction is only slightly higher than in the TP direction (i.e., $\sigma_{TP}/\sigma_{IP} < 1$), suggesting that proton transport is essentially isotropic. The calculated activation energies ($E_a$s) of Nafion® 212 in TP and IP directions are very similar, as shown in Supplementary Table 3. However, owing to the highly aligned TP-oriented PC channels of MM-45PC, a significantly higher proton conductivity in the TP direction than in the IP direction is observed, confirming anisotropic conductivity ($\sigma_{TP}/\sigma_{IP} = 2-3$ at a given temperature). The slope of the temperature-dependent conductivity line is lower for the TP direction than the IP direction, indicating that proton transport in the TP direction has lower activation energy. Especially, the $E_a$ value for MM-45PC-TP is 8.70 kJ mol$^{-1}$, which is even lower than the 11.2 kJ mol$^{-1}$ for pure hydrated PWA[63], reflecting a highly optimized proton channel structure of the MM-45PC membrane in TP direction.

The membrane stability is tested ex situ at 95 °C in water. As reported previously, without measures to stabilize embedded PWA, the simple incorporation of PWA in polymer matrix results in a rapid 96% loss at 80 °C in water within 30 days[24]. Surprisingly, MM-45PC displays strong stability for proton conductivity in both TP and IP directions, as shown in Fig. 9b, where no decrease is found at 95 °C in water for 30 days. The results listed in Supplementary Table 4 show that no PC has leached out from the MM-45PC membrane, as the IEC titration values are the same within experimental error before and after the stability challenge test. Moreover, these IEC values are very close to the theoretical value, proving that a stable complex between CP4VP and PWA occurs without loss of active protons. To the authors' knowledge, this work achieves the most robust retention of PWA filler in polymer matrix under the most severe conditions, as compared with related studies (Supplementary Table 5).

Supplementary Figure 3 shows that both water uptake and dimensional swelling of MM-45PC, largely governed by the PSf matrix in which the proton-conducting channels are embedded, are in an acceptable range for PEM. The magnetically induced phase-separated morphology allows high proton conductivity preferentially in the TP direction with a low overall level of membrane hydration, thus mitigating water uptake and dimensional swelling[64,65]. The larger IP swelling is possibly related with the orientation of CP4VP molecular chains in TP, which took place together with the induced TP movement and arrangement of the CP4VP/PWA complexes during the magnetic solution casting process.

**Single fuel cell performance**. Since in situ PEMFC performance data are very sensitive to membrane thickness, we have carefully selected three MM-45PC samples, for which the membrane cross-sectional scanning electron microscopy (SEM) images are provided in Supplementary Figure 4. All samples have very similar thickness in the range of 51–54 μm, very close to the 50.8 μm of Nafion® 212, which ensures data comparability in the in situ PEMFC test. Figure 10a and b show the single-cell PEMFC polarization curves and durability tests based on sample 1, carried out at 95 °C under 100% RH. Due to the outstanding TP proton conductivity of MM-45PC, the associated PEMFC exhibits a high maximum power density up to 1107 mW cm$^{-2}$, 172% that of a PEMFC fabricated similarly with Nafion® 212 (Fig. 10a). Supplementary Figure 5 shows comparative in situ electrochemical impedance spectra (EIS) of PEMFCs based on MM-45PC and Nafion® 212 at their power density maxima, which further

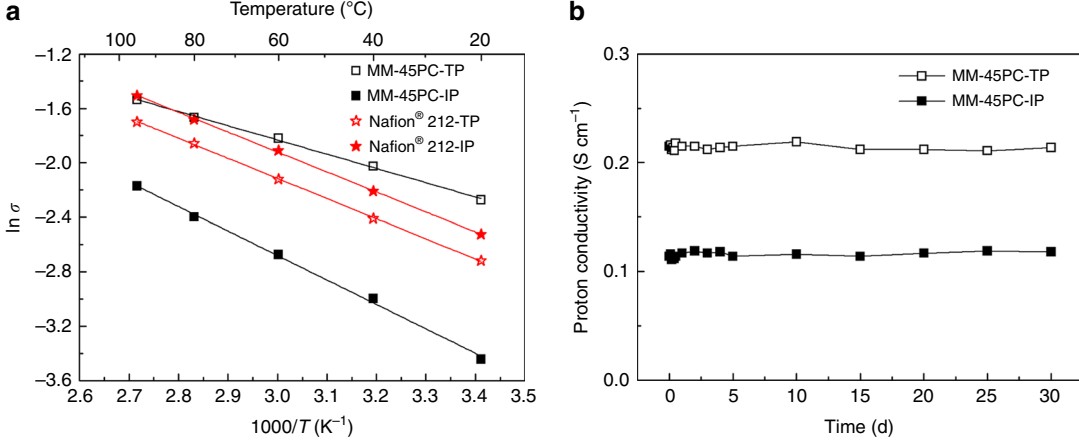

**Fig. 9** Proton conductivities of hydrated membranes measured in water. **a** Temperature-dependent TP and IP proton conductivities of MM-45PC and Nafion® 212. Compared with Nafion® 212, MM-45PC displays strongly anisotropic proton conductivity, where the TP proton conductivity is much higher than the IP conductivity, and the activation energy is lower than Nafion® 212 and MM-45PC-IP. **b** Long-term stability tests for TP and IP proton conductivities of MM-45PC at 95 °C in water. No reduction is observed within 30 days. TP: through-plane; IP: in-plane; MM-45PC: magnetic-cast membrane with 45 wt% loading of proton-conducting components. Source data are provided as a Source Data file

confirms that the higher power density of MM-45PC derives mainly from the pronounced membrane TP proton conductivity. To investigate the feasibility of MM-45PC in practical PEMFC applications, a PEMFC durability test is implemented with a constant voltage of 0.7 V. Only a small decline (4.1%) in current density occurs after 32 days, while the Nafion® 212 PEMFC incurs more than a 40% loss after 20 days (Fig. 10b). Moreover, at the conclusion of the durability tests and recovery of the membranes from the PEMFCs, MM-45PC shows no visible signs of deterioration, retaining its blue color and original shape, but Nafion® 212 shows visible and severe deterioration with a black-brown, uneven, and opaque appearance of the fractured membrane, as shown in Supplementary Figure 6. The high power density achieved combined with notable durability performance not normally encountered with hydrocarbon-based PEMs suggests that the approach used here for MM-45PC has good potential for PEMFC applications.

The 95 °C and 100% RH fuel cell operating conditions are effective in determining PWA leakage from the PEM (sample 1), but are insufficient to evaluate the overall in situ membrane durability under more aggressive PEMFC conditions which might be encountered. To demonstrate more comprehensive in situ durability of MM-45PC, chemical accelerated stress test (AST) and mechanical AST are implemented with sample 2 and sample 3, respectively.

The chemical AST is accomplished by an open circuit voltage (OCV) hold at 90 °C and 30% RH, according to the US Department of Energy suggested protocols[40,66]. As shown in Fig. 10c, after a 32-day test, the MM-45PC PEMFC displays only ~1.0% OCV loss (i.e., 13 μV h$^{-1}$), which is even less than the current density loss (4.1%) in the constant voltage test. This suggests that ferrocyanide could play an important role in helping to protect against chemical degradation. To further support this conclusion, the XPS narrow scans for Fe in MM-45PC before and after chemical AST are shown in Supplementary Figure 7. Compared with the appearance of the single Fe (II) species present in ferrocyanide before the chemical AST, the MM-45PC membrane after chemical AST shows the presence of Fe (III) as ferricyanide, indicating the redox reaction between the ferrocyanide and ferricyanide couple occurs during the OCV hold. The coexistence of ferrocyanide and ferricyanide after the long-term OCV hold reveals a rapid redox cycle without a dominant one-

way reaction, which provides highly recyclable radical scavenging. In contrast, without the presence of a radical scavenger, Nafion® 212 PEMFC exhibits a 49% OCV loss within 15 days, before catastrophic membrane failure after 17 days. The more rapid and severe chemical AST degradation of Nafion® 212 than in the constant voltage test highlights the advantages of the protective effects of the ferrocyanide groups against radical attack. It is worth noting that the 13 μV h$^{-1}$ rate of OCV loss for MM-45PC PEMFC is the lowest value reported under such aggressive conditions (90 °C/30% RH, H$_2$/O$_2$), as compared in Supplementary Table 5. The results of the chemical AST support our proposed hypothesis in the Introduction, that the ferrocyanide redox couple with strongly negative charge improves the long-term chemical stability of PEM against radical attack. This understanding will be helpful for further studies about long-term stability of PEMs and other fields where the mitigation of radical attack is important.

It appears counterintuitive that Fe$^{2+}$ is used as a degradation accelerator in Fenton's reagent for oxidative stability tests of PEMs, but the ferrocyanide–ferricyanide redox couple leads to an improvement in membrane chemical durability. To elucidate this, well-established radical formation/consumption reactions between Fe$^{2+}$ and H$_2$O$_2$ are listed below[48,49]:

$$\text{Reaction 1}: \text{H}_2\text{O}_2 + \text{Fe}^{2+} \rightarrow \text{HO}^{\bullet} + \text{OH}^- + \text{Fe}^{3+}$$
$$\text{Reaction 2}: \text{Fe}^{2+} + \text{HO}^{\bullet} \rightarrow \text{Fe}^{3+} + \text{OH}^-$$
$$\text{Reaction 3}: \text{H}_2\text{O}_2 + \text{HO}^{\bullet} \rightarrow + \text{HO}_2^{\bullet} + \text{H}_2\text{O}$$
$$\text{Reaction 4}: \text{Fe}^{2+} + \text{HO}_2^{\bullet} \rightarrow \text{Fe}^{3+} + \text{HO}_2^-$$
$$\text{Reaction 5}: \text{Fe}^{3+} + \text{HO}_2^{\bullet} \rightarrow \text{Fe}^{2+} + \text{H}^+ + \text{O}_2$$

Reaction 1 between Fe$^{2+}$ and H$_2$O$_2$ generates radicals, while reactions 2, 4, and 5 with Fe$^{2+}$/Fe$^{3+}$ consume radicals. In addition, there is a redox cycle between Fe$^{2+}$ and Fe$^{3+}$, making these reactions renewable. In the Introduction section, we assert that both OH$^{\bullet}$ and OOH$^{\bullet}$ are electrophilic because they have unpaired electrons[53,54], whose electrophilicity masks the nucleophilicity of the unshared electron pairs on oxygen. However, with the absence of any unpaired electron in H$_2$O$_2$, the nucleophilicity of the unshared electron pairs on oxygen becomes dominant[67,68]. Therefore, positively charged Fe$^{2+}$ would react more readily with nucleophilic H$_2$O$_2$ than with electrophilic OH$^{\bullet}$ and OOH$^{\bullet}$. Thus

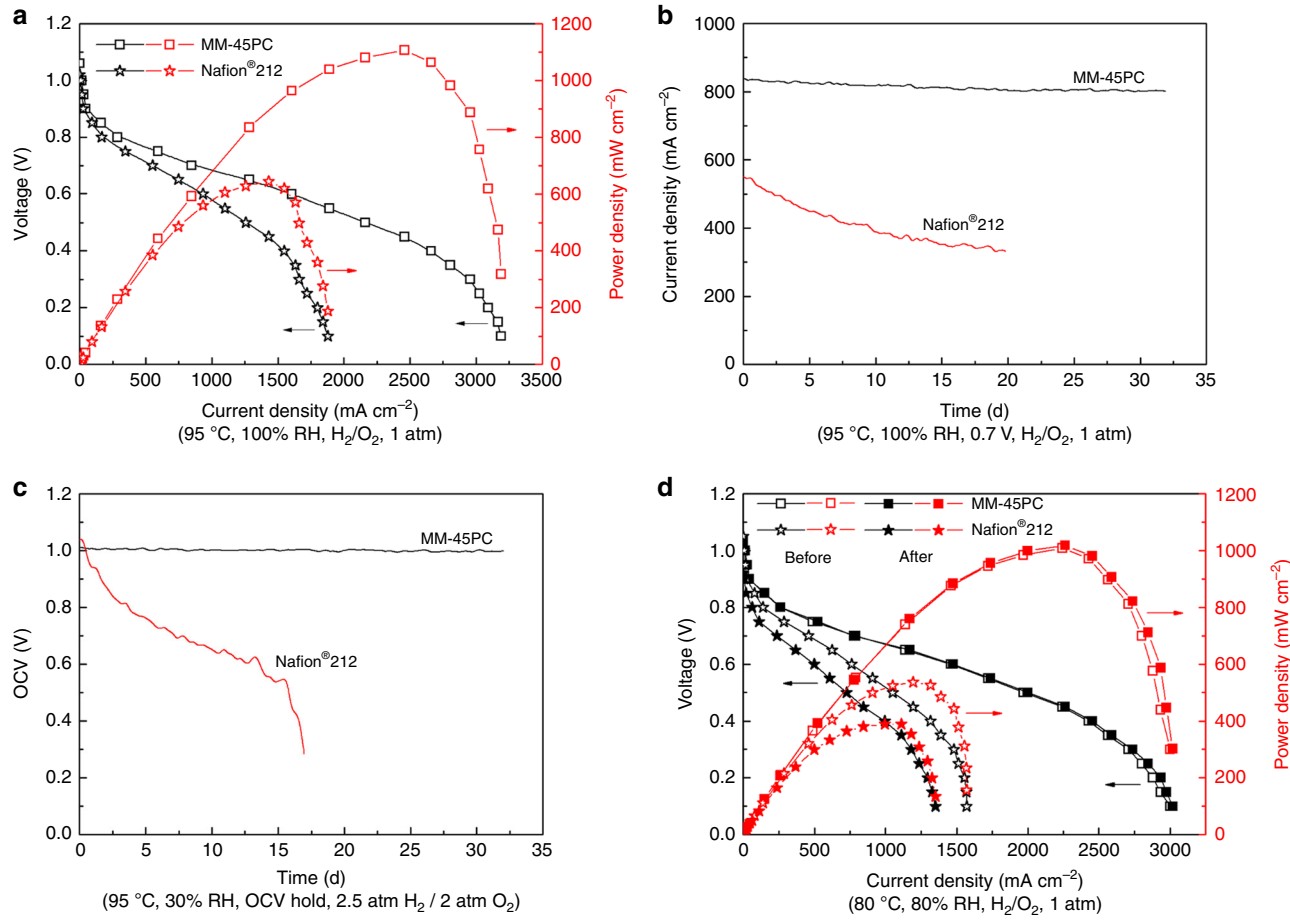

**Fig. 10** Comparative performance and durability. PEMFC performance and durability derived from MM-45PC and Nafion® 212 are shown. **a** Single-cell polarization curves for PEMFCs operated at 95 °C under 100% RH. Compared with Nafion® 212, MM-45PC shows a significantly higher maximum power density. **b** Long-term durability of PEMFC at 95 °C under 100% RH for 32 days, with a constant voltage of 0.7 V. The MM-45PC PEMFC shows only a small decline of 4.1% in current density, much lower than the 40% loss of Nafion® 212 over 20 days. **c** Chemical AST of an OCV hold at 90 °C under 30% RH. The MM-45PC PEMFC shows a decline of 1.0% OCV for 32 days, reflecting the efficient radical scavenging ability of ferrocyanide, while Nafion® 212 PEMFC exhibits catastrophic failure within 17 days. **d** Polarization curves before and after mechanical AST at 80 °C under 80% RH. The almost overlapped polarization curves of MM-45PC before and after mechanical AST indicate a higher mechanical stability compared with Nafion® 212. PEMFC: proton exchange membrane fuel cell; MM-45PC: magnetic-cast membrane with 45 wt% loading of proton-conducting components; OCV: open circuit voltage; AST: accelerated stress test; RH: relative humidity. Source data are provided as a Source Data file

$Fe^{2+}$ is chiefly a radical generator (reaction 1) rather than a radical scavenger, leaving surplus free radicals to attack other structural portions of PEM. The inverse situation occurs with the negatively charged ferrocyanide, that is, it preferentially reacts with OH• and OOH• (reaction 2, 4, and 5) rather than with $H_2O_2$. As a result, compared with $Fe^{2+}/Fe^{3+}$, the ferrocyanide/ ferricyanide redox couple are inclined to be radical scavengers, thereby protecting the PEM structure from radical attack and chemical degradation.

The mechanical AST is carried out at 80 °C for 25,000 RH 60 s cycles, with each cycle switching between 100% (30 s) and 0% RH (30 s) at both the anode and cathode, according to the US Department of Energy suggested protocols[40,66]. Different from the purpose of demonstrating PWA retention in the PEM in a hot water environment (sample 1), the polarization curves before and after mechanical AST are obtained and compared at 80 °C/80% RH. The reduced humidity tests at 80% RH are more industry-relevant operating conditions. Figure 10d shows that before the mechanical AST, the power density of MM-45PC is much higher than that of Nafion® 212 (~188%), and even higher than the 172% value of sample 1 at 95 °C/100% RH. After the mechanical AST,

the polarization curve of MM-45PC is almost unchanged, but Nafion® 212 displays an obvious reduction of ~28% loss in power density. The higher stability from the mechanical AST of MM-45PC compared with Nafion® 212 is most likely related to the tough non-ionic PSf matrix, which is quantitatively demonstrated by the ex situ membrane mechanical properties (stress–strain curve). Supplementary Figure 8 shows that MM-45PC has a much higher elastic modulus (initial slope) and tensile strength than Nafion® 212. Although the breaking elongation of Nafion® 212 is much higher than that of MM-45PC, this parameter is less relevant, because the in situ membrane dimensional change is to a relatively low extent.

The aforementioned tests provide evidence of the superior in situ durability of MM-45PC. Since the AST operating conditions are inherently more aggressive for PEMs than the constant voltage test, it is reasonable to infer that the chemical degradation and mechanical degradation of sample 1 are even lower than those of samples 2 and 3. However, the constant voltage durability test at 95 °C/100% RH (sample 1, Fig. 10b) shows a higher percentage performance loss. Excluding slight differences between membrane samples, the small loss in sample

1 is most probably caused by catalyst migration (Supplementary Figure 6), which is more likely to occur during the continuous test under high RH (100%).

## Discussion

This research is an effective example to simultaneously achieve the TP alignment of short-pathway proton channels and the stable incorporation of PWA in PEMs, thereby providing high TP proton conductivity (closely associated with PEMFC output) and membrane stability. A polymer (CP4VP) capable of strongly interacting with PWA to form a paramagnetic complex and a magnetic field are two prerequisites to achieve this. Uncomplexed CP4VP, containing a ferrocyanide group, is proton-conducting itself, and strongly proton-conducting PWA complexed with this polymer is stabilized and rendered water insoluble, contributing to both high proton conductivity and the stability of PWA in the PSf membrane matrix. Moreover, the redox cycle of ferrocyanide-ferricyanide in CP4VP continually consumes radicals generated during the PEMFC operation, significantly limiting the chemical degradation of the PEM. The opposite influences between ferrocyanide and $Fe^{2+}$ on membrane chemical degradation suggest that the appropriate deployment of redox materials with negative charge could be a promising direction for improving the durability of PEMs, and possibly beneficial in other areas where the suppression of radical degradation is required. The magnetic field has a dual function. First, it induces a stable paramagnetic complex between CP4VP and PWA. Second, the magnetic field aligns the PWA bound to the polymer chains in the TP direction of the field, creating short-pathway highly proton-conducting channels. While the present work utilizes a strong magnetic field to maximize the contrast of the distinct TP channel alignment versus random PWA incorporation, we have also observed a degree of TP channel alignment using weaker magnetic fields, although to a lesser extent. We anticipate optimizing casting conditions by controlling solution viscosity to afford more chain flexibility, which should improve TP alignment at lower field strengths. The assembled PEMFC exhibits forceful power density and durability under conditions of 95 °C and 100% RH, easily surpassing Nafion® 212. To clearly show the advantages of our work, we list some results of related PEM data in Supplementary Table 5 for comparison. Moreover, from an industrial standpoint, the starting materials used in this study are inexpensive, commercially available and simple to scale up. Compared with state-of-the-art Nafion® 212, MM-45PC shows a number of improved properties in chemical/mechanical stability and PEMFC performance under low RH. Perhaps the main challenge of this composite PEM toward commercialization will be to determine the efficacy of lower magnetic field strengths on alignment, and also the residence time needed during the casting process.

This work pioneers a feasible approach to ion conductive magnetically controllable materials, other than metallic oxides used previously, to obtain channels with efficient TP alignment in membranes. One advantage of this approach is that the low-dimensional swelling of the PEM derive largely from the hydrophobic non-conductive PSf matrix in which the aligned proton conductive channels are embedded, thereby circumventing the commonly encountered problem of excessive dimensional swelling in highly conductive PEMs. We envisage a wide variety of polymer matrices, both non-conductive and proton conducting, can be employed to embed the TP-aligned channels. The approach opens up a wide variety of other possible systems beyond membranes with aligned proton-conducting channels, such as conductive metal organic frameworks and covalent organic frameworks. We also envisage that there are many other applications requiring oriented mass transfer, including battery diaphragms and reverse-osmosis membranes, where magnetic alignment may play a crucial role in

constructing short-pathway efficient mass transfer channels to enhance performance. It is also gratifying to discover a previously unknown magnetically induced strategy for heteropoly blue formation, which may be instructive for other research in this area.

A summary of the scientific achievements of our work leading to significant performance improvements are (a) predominately TP proton channels using magnetic alignment, (b) the highest TP proton conductivity reported among similar systems, (c) a greater stability to being leached out with water than other heteropoly acid-based PEMs, (d) inhibition of free radical attack during PEMFC operation by the ferrocyanide–ferricyanide redox cycle, (e) decoupling of mechanical stability and highly conductive PEMs, and (f) a much higher fuel cell performance than the industry standard Nafion® 212 membrane.

## Methods

**Materials**. P4VP ($M_w$ ~ 60,000), PWA (99.9%), and 15-crown-5 (98%) were purchased from Sigma-Aldrich, China. SPCAF (95%) was bought from Tokyo Chemical Industry, Japan. PSf ($M_w$ ~ 60000) was obtained from Acros, China. Reagent grade methanol (99%), isopropanol (99%), and dimethylformamide (DMF) (99%), benzene (99%), aqueous HCl (1 M), aqueous NaOH (10.0 mM), and phenolphthalein (0.2 wt% in ethanol) were obtained from Aladdin, China. All the materials were used as received.

**Synthesis of coordinated polymer**. A solution of SPCAF (10.9 g, 40 mmol) and 15-crown-5 (26.4 g, 120 mmol) in water (50 mL) was added dropwise to a magnetically stirred solution of P4VP (1.05 g, 10 mmol) in methanol (50 mL). The mixed liquid was sealed and stirred for 72 h at 40 °C. The solution was allowed to cool to room temperature, filtered through a syringe filter (polytetrafluoroethylene (PTFE) with a pore size of 0.45 μm), mixed with 100 mL of 1 M HCl in an ice bath, and then very slowly decanted into isopropanol to precipitate CP4VP. The crude polymer product was re-dissolved in 100 mL of 1 M HCl in an ice bath, stirred for 1 h, and reprecipitated in isopropanol. This procedure was carried out three times to obtain purified fully acidified polymer. After being dried at room temperature for 24 h in a vacuum oven, the resulting green-yellow colored CP4VP polymer was stored in a desiccator.

**Fabrication of composite membrane**. Membrane casting solutions were prepared by dissolving CP4VP (0.045 g), PWA (0.135 g) and PSf (0.22 g) in DMF (5.0 g) and water (1.0 g) contained in vials, which were agitated on a test tube gyrator (Thermo Fisher Scientific, USA) for 12 h. After filtration using a syringe filter (0.45 μm PTFE) and degassing to remove bubbles, the solutions were poured into a circular membrane casting dish (5 cm diameter) and the solvent was allowed to evaporate at 80 °C with or without magnetic field (35 tesla water-cooled magnet). The resulting membranes were then immersed in 1 M HCl for 24 h and washed repeatedly by deionized (DI) water until the pH value was 7.0. Five membrane samples for each formulation were used to ensure data reproducibility. Membrane thickness was in the range of 50–60 μm.

**Characterization**. XPS were measured by an Escalab 250 (Thermo Fisher Scientific, USA) instrument that employs Al Kα radiation. FTIR were conducted on a Nicolet iS10 system (Thermo Fisher Scientific, USA) in the range of 4000–700 cm$^{-1}$. TGA was characterized by a Q500 system (TA Instruments, USA) under nitrogen atmosphere with a heating rate of 10 °C min$^{-1}$. DSC measurement was carried out on a Q2000 instrument (TA Instruments, USA) under nitrogen atmosphere using a heating rate of 10 °C min$^{-1}$. Elemental analysis measurement was conducted using a Vario Micro Cube (Elementar, Germany). UV–Vis spectrum analysis was done using a Lambda 750 spectrophotometer (Perkin Elmer, USA). The magnetic property was tested using VSM on a Squid-VSM system (Quantum Design, USA) at 80 °C. Cross-sectional membrane samples for TEM (Tecnai G$^2$ F20 field emission TEM, FEI, USA) were embedded in epoxy resin and solidified at 100 °C for 2 h, and then cut on an EMUC6 ultramicrotome (Leica, Germany). Elements W, Fe, and S were detected and mapped by an Enfinium 977 EELS spectrometer (Gatan, USA), and the mapping time was 2 s. The particle size of PWA was estimated by DLS technique on a Zetasizer nano ZS90 apparatus (Malvern Instruments, UK). Cross-sectional membrane samples for SEM (SU8010 field emission SEM, Hitachi, Japan) were freeze-fractured in liquid nitrogen and vacuum sputtered with a thin layer of gold.

**Proton conductivity and electron conductivity**. Membrane samples were equilibrated in DI water for 24 h (for long-term stability test, the time was longer) before they were assembled in a custom-built, open-frame, two-electrode clamp. An autolab-PGSTAT302N electrochemical workstation (Metrohm, Switzerland) was employed for both TP and IP total conductivity (sum of proton and electron conductivity) measurement using alternating current (AC) impedance technique, or electron conductivity measurement using direct current (DC) resistance

technique[69]. The membrane total conductivity, $\sigma_{\text{total}}$, was calculated using Eq. 1:

$$\sigma_{\text{total}} = \frac{l}{R_{\text{AC}} \cdot S} \tag{1}$$

and electron conductivity, $\sigma_e$, was calculated using Eq. 2:

$$\sigma_e = \frac{l}{R_{\text{DC}} \cdot S} \tag{2}$$

where $l$, $R_{\text{AC}}$, $R_{\text{DC}}$, and $S$ are the distance between two electrodes, the resistance of AC impedance test, the resistance of DC resistance test, and the section area of the membrane, respectively. Averaged values were obtained from five samples. Strictly speaking, the proton conductivity, $\sigma$, should be calculated from $\sigma = \sigma_{\text{total}} - \sigma_e$, but for most cases, the electron conductivity of PEM is negligible, so the $\sigma_{\text{total}}$ obtained by AC impedance test is usually directly used as the proton conductivity $\sigma$.

**Water uptake and dimensional swelling**. Membranes were equilibrated in DI water for 24 h before the measurements of water uptake and swelling.

For water uptake measurement, membranes were immediately blotted to remove the surface water and weighed to obtain the wet mass $W_{\text{wet}}$. After being dried in a vacuum oven at 60 °C for 24 h, the membranes were weighed again to obtain the dry mass $W_{\text{dry}}$. The water uptake is calculated using Eq. 3:

$$\text{Water uptake (wt\%)} = \frac{W_{\text{wet}} - W_{\text{dry}}}{W_{\text{dry}}} \times 100. \tag{3}$$

For dimensional swelling measurements, the length, width, or thickness of the hydrated membranes were measured and noted as $L_{\text{wet}}$. After being dried in a vacuum oven at 60 °C for 24 h, the membrane length, width, or thickness were measured again and noted as $L_{\text{dry}}$. The dimensional swelling is obtained by using Eq. 4:

$$\text{Swelling (\%)} = \frac{L_{\text{wet}} - L_{\text{dry}}}{L_{\text{dry}}} \times 100. \tag{4}$$

The increase in membrane thickness is the TP dimensional swelling and the increase in length or width is the IP dimensional swelling. Averaged values were obtained from five samples.

**Ion exchange capacity**. The IEC of CP4VP and membranes were obtained by titration. A measured amount (about 200 mg) of dried sample was soaked in 15.00 mL 5 M aqueous NaCl for 24 h to release H$^+$. Then, 10.00 mL of the solution was pipetted into a conical flask (to avoid the decomposition of PWA in alkaline condition) and titrated by a standard 10.0 mM aqueous NaOH with phenolphthalein as the indicator. The IEC was calculated from Eq. 5:

$$\text{IEC} \left( \text{mmol g}^{-1} \right) = \frac{c_{\text{NaOH}} \cdot v_{\text{NaOH}}}{m} \times \frac{15}{10} \times 1000, \tag{5}$$

where $c_{\text{NaOH}}$ and $v_{\text{NaOH}}$ are the concentration and the used volume of the NaOH solution, and $m$ is the mass of the sample. Averaged values were obtained from five samples.

**Membrane electrode assembly preparation and fuel cell tests**. Commercial Pt/C (60 wt% Pt, Johnson Matthey, England) was used as the catalyst for both anode and cathode. The catalysts were dispersed in Nafion® binder (Nafion® D521 dispersion, Alfa Aesar, China), where the mass ratio of Nafion® to the catalyst was 20 wt%. The resulting dispersion was sprayed by an air gun (Iwata, Japan) onto carbon paper (Toray 250, Japan) to achieve 0.4 mg cm$^{-2}$ catalyst loadings (0.24 mg Pt cm$^{-2}$) on both anode and cathode with an effective area of 4 cm$^2$. Membrane electrode assembly was fabricated from the anode–membrane–cathode sandwich by hot press under a pressure of 4.0 MPa at 120 °C for 3 min. Single-cell PEMFC tests were conducted on a Fuel Cell Test System 850e (Scribner Associates, USA). The hydrogen and oxygen were supplied at selected RH values to the anode and cathode at flow rates of 120 and 160 sccm (standard atmospheric pressure without back pressure), respectively. The cell was activated at the selected test temperature for 3 h before the polarization curves were recorded. A 10 mV AC signal was included to the PEMFC to acquire EIS information, with the scanning frequency varied from 0.1 to 10$^5$ Hz.

**Accelerated stress tests**. The chemical ASTs were performed by an OCV hold at 90 °C, with 30% RH hydrogen and 30% RH oxygen (2.5 atm hydrogen and 2.0 atm oxygen, without back pressure) for the anode and cathode, respectively. The mechanical AST were carried out at 80 °C for 25,000 RH 60 s cycles, with each cycle switching between 100% (30 s) and 0% RH (30 s) at both the anode and cathode, according to the US Department of Energy suggested protocols[40,66].

**Ex situ membrane mechanical property**. Dumbbell-shaped samples were cut from membranes by a CP-25 sheet-punching machine (Creator, China). These

samples were placed in a ZP (H) 32 chamber (Cincinnati Sub-Zero, USA) at 23 °C and 50% RH for 48 h before test, according to ASTM D882. The stress–strain curves were obtained by the tensile test on a Q800 system (TA Instruments, USA), at the speed of 5 mm min$^{-1}$. The gauge length and the width of the samples were 30 and 5 mm, respectively.

## Data availability

The source data underlying Figs. 3, 4, 5, 6, 7, 9 and 10 and Supplementary Figs. 1, 2, 3, 5, 7 and 8 have been deposited in the figshare repository at https://figshare.com/ (https://doi.org/10.6084/m9.figshare.7542185).

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

## Acknowledgements

We thank the National Natural Science Foundation of China (21875161). We also thank the National Key Technology R&D Program (2016YFB0303300), the Natural Science Foundation of Tianjin (17JCZDJC31000), State Key Laboratory of Engines, and State Key Laboratory of Separation Membranes and Membrane Processes (M1-201704) for financial support.

## Author contributions

M.D.G. conceived the study. M.D.G. and X.L. designed the experiments. M.D.G., Y.Y., J.Z. and X.L. wrote the manuscript. X.L., J.X. and Y.L. carried out the experiments and collected the data. X.L. and Y.L. prepared the data graphs. M.D.G., Y.Y., J.Z., X.L. and W.Z. contributed to the sketches. M.D.G., Y.Y., J.Z., Y.Q., K.J., Q.D., B.C., X.Z., J.L. and X.L. discussed the results. All authors commented on the manuscript.

## Additional information

**Competing interests:** The authors declare no competing interests.

