## [Peer Review File · Nature Communications]

Reviewers' Comments:

Reviewer #1:

Remarks to the Author:

In this paper, the authors investigated a composite PEM containing TP-aligned proton channels with PWA. But this conclusion is not fully supported by the experimental data and is not logically organized. As a general comment, the novelty and originality of the article do not meet the expectation of the Nature communications in scientific quality and impact. For the reason, I recommend the paper should be rejected. However, I believe that consideration of these comments will lead to an improved report that better illustrates the key concepts and conclusions.

* There are too many unnecessary references. This makes it practically difficult to check the necessary information.

* Please refrain from using subjective terms such as 'super-stable PWA'.

* If the stability of this electrolyte material is ensured, please explain what caused the reduction in the stability of this fuel cell.

* The motivation of this study is unclear.

* In order to highlight the originality and features of this work, please explain specifically and clearly why the stability of this material has improved and compare this with other references in figure10 to see how well the MM-45PC is performing.

Reviewer #2:

Remarks to the Author:

This work deals with a novel approach for alignment of proton conducting channels in a membrane of a PEM fuel cell. The material investigated is a ferrocyanide-polymer and a phosphotungsten acid.

While the results are very interesting and promising, I cannot recommend this paper for publication in Nature Communications in the present form.

My main concern is related to the statement the authors make on page 4 line 78 - 80: 'With these approaches, enhanced stability of PWA composite PEMs was achieved, but long-term membrane stability tests (>30 days) under severe conditions (>90 oC in water) are not reported.'

If the only novelty in this paper are the fuel cell experiments, the novelty is too low for this journal. In this case, this work might be suited for another more fuel cell specific journal such as fuel cells.

In any case: The authors need to more carefully elaborate the difference between the novelty of this others and others.

The language needs some improvement and therefore I would recommend to recheck the manuscript by a native speaker. Also I would recommend using past tense in the manuscript more than present tense.

Very generally, the authors also show a lack of precision performing the fuel cell comments. My main, more detailed concerns on that are given below.

A few smaller remarks:

-p.3 the authors claim: '...exhibit isotropic conductivity or even unfavorable anisotropy with

lower proton conductivity in the TP direction compared with the in-plane (IP) direction, despite often having distinct phase-separated morphology with tortuous-path conductive channels...' This statement, if true, is not common sense and thus must be proven with literature sources.

- The pressure used in the fuel cell experiments is pivotal and is not given in the information. For an assessment of the performance, this must be given. Also, the fuel cell experimental conditions should be given below the figure corresponding to the measurements.
- Which accelerated did you choose and why? Where is the reference to it in the text?
- For me it is both strange and interesting, that a relatively small change in ionic conductivity (acc. to fig. 9 a) results in quite a substantial change in the the power density. Impedence spectra are needed here along with a proper discussion of its properties to interpret the results. One option would be that the membrane was thinner than reported by the authors, who state a thickness of 50 - 60 μm for all membranes. For that an image of the measured samples membrane by SEM is mandatory.

Without a proper implementation of these points I can not recommend publication.

Reviewer #3:

Remarks to the Author:

The major achievement of this paper is a scheme by which proton conducting channels are created orthogonal to the plane of the membrane. The fuel cell performance improvements achieved by this method are impressive, but they are reported 100% RH, a condition which is no longer industrially relevant. Furthermore, the authors should comment on the practicality of this system for mass production of fuel cell membranes.

Much is made of the stability of the heteropoly acid in the film. However, the authors are missing the point, chemical stability of the film is due to mitigation of radicals, the heteropoly acid chosen is a known activator of radicals. The data shown for stability is under wet conditions where radicals do not predominate, how would this film survive in an OCV accelerated stress test under dry conditions at 90oC? Interestingly Fe is strongly implicated in PEM fuel cell degradation, can the authors comment on the stability of their iron based polymer?

If the formation of heteropoly blues is not reversible then then the membrane will conduct electrons and will be worthless in a fuel cell, what is the mechanism of re-oxidation of the heteropoly acids?

Nafion is a trade name and should was have the registered mark superscripted after Nafion.

Reviewer #1

R1-1: In this paper, the authors investigated a composite PEM containing TP-aligned proton channels with PWA. But this conclusion is not fully supported by the experimental data and is not logically organized. As a general comment, the novelty and originality of the article do not meet the expectation of the *Nature Communications* in scientific quality and impact. For the reason, I recommend the paper should be rejected. However, I believe that consideration of these comments will lead to an improved report that better illustrates the key concepts and conclusions.

R1-1 response: We demonstrate a novel approach to prepare through-plane aligned proton-conducting membranes composed of heteropoly acids tethered to polymers. The novelty and originality of our work leading to significant performance improvements are (a) predominately through-plane proton channels through magnetic alignment, (b) the highest through-plane proton conductivity reported among similar systems (we had included a comparative table of performance in Supplementary Table 5), (c) a much greater stability to being leached out with water than any other system, by heating the PEM in water for an extended time, which no other heteropoly acid systems could endure, (d) inhibition of free radical attack during PEMFC operation by the presence of the ferrocyanide-ferricyanide redox cycle, (e) decoupling of mechanical stability and highly conductive PEMs, and (f) a much higher fuel cell performance than the industry standard Nafion[®] 212 membrane. The novelty and originality of our approach is markedly different from any other approach reported. We have revised the Introduction and Discussion section in the main text to state our point more clearly.

R1-2 There are too many unnecessary references. This makes it practically difficult to check the necessary information.

R1-2 response: We believe the majority of the references are relevant to a scholarly article of the calibre of *Nature* publication series. We have rechecked the bibliography and cited the appropriate references.

R1-3 Please refrain from using subjective terms such as 'super-stable PWA'.

R1-3 response: We accept this criticism of using inappropriate subjective expressions such as 'super-stable PWA' and 'ultra-strong'. We have replaced instances of these expressions with more appropriate and moderate wording.

R1-4 If the stability of this electrolyte material is ensured, please explain what caused the reduction in the stability of this fuel cell.

R1-4 response: Operating fuel cells are dynamic complex multi-component systems. The observed slight reduction in PEMFC performance over 32 days (Fig. 10b) could be caused by multiple factors, but most likely by catalyst migration during long-term

100% RH operation (Supplementary Fig. 6). It is noteworthy that a comparative PEMFC fabricated from industry-standard Nafion[®] 212 did not exhibit similarly high performance under these conditions, and in fact, completely failed after 20 days. We have added related discussion about this point in the revised main text (page 26).

R1-5 The motivation of this study is unclear.

RI-5 response: The motivation of this study is the fabrication of a stable PEM with shorter membrane through-plane conducting pathways having faster proton transport than typical PEMs with isotropic conductivity. The means to achieve this is by constructing a highly oriented structure that particularly promotes proton transport in the membrane in the through-plane direction. By embedding the conducting channels in a mechanically strong non-conductive membrane matrix, we decouple the commonly observed phenomenon of poor mechanical properties in highly conductive PEMs. We believe we expressed our motivation clearly in the Introduction.

R1-6 In order to highlight the originality and features of this work, please explain specifically and clearly why the stability of this material has improved and compare this with other references in figure10 to see how well the MM-45PC is performing.

R1-6 response: We had already included a comparison of our results with other related work (Supplementary Table 5). The stability of our PEM originates from the stable incorporation of PWA (Fig. 3, Fig. 9b and Supplementary Table 4), the renewable radical scavenging activity of ferrocyanide groups (Supplementary Fig. 7) and the strong mechanical properties of the tough PSf matrix into which the conducting channels are embedded (Supplementary Fig. 8).

Reviewer #2

This work deals with a novel approach for alignment of proton conducting channels in a membrane of a PEM fuel cell. The material investigated is a ferrocyanide-polymer and a phosphotungsten acid.

While the results are very interesting and promising, I cannot recommend this paper for publication in Nature Communications in the present form.

R2-1 My main concern is related to the statement the authors make on page 4 line 78 - 80:

'With these approaches, enhanced stability of PWA composite PEMs was achieved, but long-term membrane stability tests (>30 days) under severe conditions (>90 oC in water) are not reported.'

If the only novelty in this paper are the fuel cell experiments, the novelty is too low for this journal. In this case, this work might be suited for another more fuel cell specific journal such as fuel cells.

In any case: The authors need to more carefully elaborate the difference between the novelty of this others and others.

R2-1 response: We believe that our work has at least six elements of novelty and significant performance improvements over existing work. Please see our response to the comments of Reviewer 1.

R2-2 The language needs some improvement and therefore I would recommend to recheck the manuscript by a native speaker. Also I would recommend using past tense in the manuscript more than present tense.

R2-2 response: The corresponding author, a native English speaker, edited the original manuscript and has carefully rechecked the revised manuscript. We have since communicated with the Editor about the language requirements in *Nature* series journals, particularly about the tense. The Editor commented that the Abstract and Introduction must use the present tense, whereas the remainder of the manuscript can be either in the present or past tense.

Very generally, the authors also show a lack of precision performing the fuel cell comments. My main, more detailed concerns on that are given below.

A few smaller remarks:

R2-3 -p.3 the authors claim: '...exhibit isotropic conductivity or even unfavorable anisotropy with lower proton conductivity in the TP direction compared with the in-plane (IP) direction, despite often having distinct phase-separated morphology with tortuous-path conductive channels...' This statement, if true, is not common sense and thus must be proven with literature sources.

R2-3 response: This statement has been supported by references 1-5 in the original main text. References 1-5 in the original main text were not only used to raise the concepts of perfluorocarbon-based PEM and hydrocarbon-based PEM, but also the related studies to present the “isotropic conductivity or even unfavorable anisotropy” in perfluorocarbon-based PEM and hydrocarbon-based PEM. We cite more references in the revised main text to support the generality of this statement: 1-4 for perfluorocarbon-based PEM, and 5-8 for hydrocarbon-based PEM. We now cite references 1-8 in the end of the statement (revised main text, page 3).

We believe this statement is supported by these cited references.

R2-4-The pressure used in the fuel cell experiments is pivotal and is not given in the information. For an assessment of the performance, this must be given. Also, the fuel cell experimental conditions should be given below the figure corresponding to the measurements.

R2-4 response: We thank the reviewer for pointing out this omission. The pressure used in polarization curves and constant voltage experiments was standard atmospheric pressure without back pressure, and the pressure used in chemical and

mechanical ASTs were according to the suggested protocols by the U.S. Department of Energy (main text, page 23, 25 and 35-36). This important information is now included in the main text, page 35-36.

Furthermore, the fuel cell experimental conditions are now given below each subfigure of Fig. 10.

R2-5-Which accelerated did you choose and why? Where is the reference to it in the text?

R2-5 response: We had already included fuel cell performance over a 32-day period, which is perhaps longer than a majority of work reported in the literature. However, we have now included a chemical accelerated stress test (AST) and mechanical AST in the revised main text, (Fig. 10c and 10d, page 27, and related discussion, page 22-26), according to the suggested protocols by the U.S. Department of Energy (main text, page 23, 25 and 35-36).

R2-6 For me it is both strange and interesting, that a relatively small change in ionic conductivity (acc. to fig. 9 a) results in quite a substantial change in the the power density. Impedance spectra are needed here along with a proper discussion of its properties to interpret the results. One option would be that the membrane was thinner than reported by the authors, who state a thickness of 50 - 60 μm for all membranes. For that an image of the measured samples membrane by SEM is mandatory.

R2-6 response: We thank the reviewer for this insightful comment. The in-plane and through-plane proton conductivities of hydrated PEMs in Fig. 9a were tested at 95 °C in liquid water (18% change between MM-45PC and Nafion[®] 212), whereas the PEMFC tests in Fig. 10a were carried out at 95 °C under 100% RH (72% change between MM-45PC and Nafion[®] 212). The experimental conditions were different and thus the data are not directly correlated. In order to draw a more direct comparison between membrane proton conductivity data and PEMFC power density, we performed additional experimental work to obtain the through-plane conductivity of MM-45PC and Nafion[®] 212 at 95 °C under 100% RH (i.e. the same conditions used in the PEMFC test). The difference in proton conductivity (89%) was higher than that in PEMFC power density (72%). Following the reviewer's comment, we re-examined the impedance spectra, and found that the difference of PEMFC performances between MM-45PC and Nafion[®] 212 derives mainly from differences in their high frequency intercepts, which are directly related to membrane proton conductivities, as shown in Supplementary Fig. 5. The change in membrane proton conductivities between liquid water and a 100% RH atmosphere are quite marked for Nafion[®] 212, which is one of the principal reasons in exploring other PEMs. The perfluorinated structure of Nafion[®] 212 is susceptible to severe membrane dehydration under water vapor atmosphere at an elevated temperature of 95 °C, without the osmotic pressure of the liquid water. Moreover, the current density of MM-45PC PEMFC is much larger than that of Nafion[®] 212 PEMFC in impedance spectra tests, which indicates faster water generation and additional humidification, so the difference in high frequency intercepts (Supplementary Fig. 5) is further increased

(140%) compared with the difference in the *ex situ* proton conductivities (89%).

We agree that all *in situ* PEMFC performance data are very sensitive to membrane thickness. Knowing this, we had carefully selected three MM-45PC samples for PEMFC and other membrane evaluation. The membrane cross-sectional SEM images are now provided in Supplementary Fig. 4. Sample 1 was used for the polarization curve and constant voltage test, sample 2 for the chemical AST and sample 3 for the mechanical AST. All three samples had very similar thickness in the range of 51~54 μm , similar with the 50.8 μm of Nafion[®] 212, and for each sample, the thickness is quite uniform along the cross section. Thus, since the membrane thickness of the membrane samples were similar with each other and that of Nafion[®] 212, we believe the *in situ* PEMFC performance is in line with the membrane proton conductivity data.

Without a proper implementation of these points I can not recommend publication.

Reviewer #3

R3-1 The major achievement of this paper is a scheme by which proton conducting channels are created orthogonal to the plane of the membrane.

R3-1 response: We thank the reviewer for the positive comment. Please see our response to the comments of Reviewer 1 on the novel and original aspects of our work.

R3-2 The fuel cell performance improvements achieved by this method are impressive, but they are reported 100% RH, a condition which is no longer industrially relevant. Furthermore, the authors should comment on the practicality of this system for mass production of fuel cell membranes.

R3-2 response: Since the retention of PWA is one of the key topics of this research, the polarization curve in Fig. 10a was carried out under 100% RH, which is the highest hydration level in PEMFC test, most severe and effective *in situ* condition to check PWA leaching out. We have carried out the polarization curves at 80 °C under 80% RH in the revised main text (Fig. 10d, page 27, and related discussion, page 25), a more industrially relevant condition, and found even higher improvement in power density compared with Nafion[®] 212.

The comments on the practicality of this system for mass production of fuel cell membranes have been added at the end of the first paragraph of the Discussion section (revised main text, page 29).

Therefore, MM-45PC has the potential for industrialization.

R3-3 Much is made of the stability of the heteropoly acid in the film. However, the authors are missing the point, chemical stability of the film is due to mitigation of radicals, the heteropoly acid chosen is a known activator of radicals. The data shown for stability is under wet conditions where radicals do not predominate, how would this film survive in an OCV accelerated stress test under dry conditions at 90°C? Interestingly Fe is strongly implicated in PEM fuel cell degradation, can the authors comment on the stability of their iron based polymer?

R3-3 response: The reviewer raises some important points, for which we are grateful. We conducted some additional experimental work to address these concerns. Chemical AST (OCV hold) was carried out at 90 °C under 30% RH, and the result was added in the revised main text (Fig. 10c, page 27, and related discussion, page 23-24). The OCV loss was ~1.0 % for 32 d (Fig 10c), and the reason for the strong chemical stability observed was also discussed (revised main text, page 24-25, and Supplementary Fig. 7).

We added some clarification about the opposite influence of Fe²⁺ on membrane chemical stability by AST (main text, page 24-25). Thus, the feasibility of CP4VP and MM-45PC as PEM materials is demonstrated.

R3-4 If the formation of heteropoly blues is not reversible then then the membrane will conduct electrons and will be worthless in a fuel cell, what is the mechanism of re-oxidation of the heteropoly acids?

R3-4 response: We appreciate the concerns of the reviewer, but our experimental PEMFC performance data indicate a high degree of stability over a 32-day period. If PWA were to undergo a reversible redox process, it would become un-tethered from CP4VP and then be soluble and detectable in water. To verify whether this occurs, we conducted some additional experimental work. A UV-Vis test (Fig. 3), proton conductivity stability (Fig. 9b) and IEC titration (Supplementary Table 4) demonstrated that leaching of PWA from the membrane was not detectable. Therefore, we believe that PWA tethered to CP4VP does not undergo a reversible redox reaction.

To verify the electron conductivity of the membrane, we did some additional experimental work. The electron conductivity of MM-45PC (see Supplementary Information “Membrane electron conductivity”) was measured by the DC resistance method. The electron conductivity of MM-45PC was very low, and much lower than that of Nafion[®] 212 (Supplementary Fig. 2). Consequently, we believe that the very low electron conductivity would not restrict the prospect of MM-45PC in a PEMFC application.

R3-5 Nafion is a trade name and should have the registered mark superscripted after Nafion.

R3-5 response: We added the registered mark after each instance of ‘Nafion’.

Reviewers' Comments:

Reviewer #1:

Remarks to the Author:

I appreciate your answer. But it is still hard to understand from this manuscript whether this manuscript satisfies the impact of the Nature communications. For example, there are many electrolytes for PEMFCs that are sufficient to be commercialized, and it is difficult to understand the superiority of this material over these state-of-the-art PEMFC electrolytes. Thus, I cannot understand the authors' standpoint and recommend the paper should be rejected.

Reviewer #2:

Remarks to the Author:

With the presented additional works the paper can now be published.

Reviewer #1

Reviewer #1 (Remarks to the Author):

R1-1: I appreciate your answer. But it is still hard to understand from this manuscript whether this manuscript satisfies the impact of the Nature communications. For example, there are many electrolytes for PEMFCs that are sufficient to be commercialized, and it is difficult to understand the superiority of this material over these state-of-the-art PEMFC electrolytes. Thus, I cannot understand the authors' standpoint and recommend the paper should be rejected.

R1-1 response: We regret that the reviewer was still not able to understand some of the improved properties of our PEM compared with state-of-the-art electrolytes, although we provided comparative data in the original and subsequent submissions.

While many non-commercial hydrocarbon polymer-based electrolytes for PEMFC are reported, the large majority of them have unsuitable properties, exhibiting problematic issues such as rapid degradation under fuel cell conditions, low proton conductivity at reduced relative humidity, or severe dimensional swelling with concurrent loss in mechanical properties. When compared with commercial state-of-the-art Nafion[®] 212, we demonstrate our PEM (MM-45PC) achieved significant progress in three aspects. First, with the radical scavenging capacity deriving from the redox of ferrocyanide-ferricyanide (Supplementary Fig. 7), the chemical stability of MM-45PC is well above Nafion[®] 212 (Fig. 10c) under the conditions tested in a fuel cell. Second, with the tough PSf matrix (Supplementary Fig. 8), the mechanical stability of MM-45PC also displays some advantage over Nafion[®] 212 (Fig. 10d). The unique structural feature of oriented proton channels embedded in tough matrix also decouples the commonly observed phenomenon of poor mechanical properties in highly conductive PEMs. Third, the relatively small difference in membrane proton conductivity in water between MM-45PC and Nafion[®] 212 (18% difference in Fig. 9a) results in a substantially higher PEMFC power density (72% difference in Fig 10a). MM-45PC overcomes the severe membrane dehydration seen in Nafion[®] 212 under water vapor atmosphere. Furthermore, the performance advantage of MM-45PC PEMFC compared with Nafion[®] 212 PEMFC is magnified under low RH operating conditions (72% higher at 95 °C/100% RH in Fig. 10a and 88% higher at 80 °C/80% RH in Fig. 10d).

Similar with Nafion[®] 212, most commercial PEMs are perfluorosulfonic acid-based materials. The proton conductive sulfonic group is susceptible to radical attack. The perfluorinated main chain has relatively weak mechanical strength and is also susceptible to membrane dehydration. To further emphasize selected improved properties of MM-45PC, we have now added a new sentence "Compared with state-of-the-art Nafion[®] 212, MM-45PC shows a number of improved properties in chemical/mechanical stability and PEMFC performance under low RH." in the Discussion section (revised main text, page 21-22).

Reviewer #2 (Remarks to the Author):

R2-1 With the presented additional works the paper can now be published.

R2-1 response: We thank the reviewer for the positive comment on our revised manuscript.